# Oncogenic Effects of HIV-1 Proteins, Mechanisms Behind

**DOI:** 10.3390/cancers13020305

**Published:** 2021-01-15

**Authors:** Maria Isaguliants, Ekaterina Bayurova, Darya Avdoshina, Alla Kondrashova, Francesca Chiodi, Joel M. Palefsky

**Affiliations:** 1Gamaleya Research Center for Epidemiology and Microbiology, 123098 Moscow, Russia; bayurova_eo@chumakovs.su (E.B.); avdoshina_dv@chumakovs.su (D.A.); 2M.P. Chumakov Federal Scientific Center for Research and Development of Immune-and-Biological Products of Russian Academy of Sciences, 108819 Moscow, Russia; kondrashova_as@chumakovs.su; 3Department of Microbiology, Tumor and Cell Biology, Karolinska Institutet, 17177 Stockholm, Sweden; francesca.chiodi@ki.se; 4Department of Research, Riga Stradins University, LV-1007 Riga, Latvia; 5Department of Medicine, University of California, San Francisco, CA 94117, USA; joel.palefsky@ucsf.edu

**Keywords:** human immunodeficiency virus type 1, epithelial cells, carcinogenicity, oxidative stress, reactive oxygen species, gp120, Tat, Nef, matrix protein p17, reverse transcriptase

## Abstract

**Simple Summary:**

People living with human immunodeficiency virus type 1 (HIV-1) (PLWH) are at increased risk of developing cancer despite successful antiretroviral therapy (ART). Here, authors suggest novel mechanism behind this phenomenon. HIV proteins, namely envelope protein gp120, accessory protein negative factor Nef, matrix protein p17, transactivator of transcription Tat and reverse transcriptase RT, are known to be oncogenic per se, to induce oxidative stress and to be released from the infected or expressing cells. These properties are proposed to underlie their capacity to affect bystander epithelial cells causing their malignant transformation, and to enhance tumorigenic potential of already transformed/cancer cells. HIV proteins can act alone or in collaboration with other known oncoproteins, specifically originating from the oncogenic human viruses such as human hepatitis B and C viruses, and human papilloma viruses of high carcinogenic risk, which cause the bulk of malignancies in people living with HIV-1 on ART.

**Abstract:**

People living with human immunodeficiency virus (HIV-1) are at increased risk of developing cancer, such as Kaposi sarcoma (KS), non-Hodgkin lymphoma (NHL), cervical cancer, and other cancers associated with chronic viral infections. Traditionally, this is linked to HIV-1-induced immune suppression with depletion of CD4+ T-helper cells, exhaustion of lymphopoiesis and lymphocyte dysfunction. However, the long-term successful implementation of antiretroviral therapy (ART) with an early start did not preclude the oncological complications, implying that HIV-1 and its antigens are directly involved in carcinogenesis and may exert their effects on the background of restored immune system even when present at extremely low levels. Experimental data indicate that HIV-1 virions and single viral antigens can enter a wide variety of cells, including epithelial. This review is focused on the effects of five viral proteins: envelope protein gp120, accessory protein negative factor Nef, matrix protein p17, transactivator of transcription Tat and reverse transcriptase RT. Gp120, Nef, p17, Tat, and RT cause oxidative stress, can be released from HIV-1-infected cells and are oncogenic. All five are in a position to affect “innocent” bystander cells, specifically, to cause the propagation of (pre)existing malignant and malignant transformation of normal epithelial cells, giving grounds to the direct carcinogenic effects of HIV-1.

## 1. Introduction

Immune suppression and related dysfunctions result in a high prevalence in people living with human immunodeficiency virus (PLWH) of HIV-1/AIDS-associated disorders, including so called AIDS-defining cancers (ADC)—Kaposi sarcoma (KS), non-Hodgkin lymphoma (NHL) and cervical cancer. In the era of antiretroviral therapy (ART), their rates have sharply declined: KS by 60–70% and NHL, by 30–50% compared to the pre-ART era. Still, the incidence of KS in PLWH remains elevated 800-fold, of NHL 10-fold and of ADC 4-fold, compared to their rates in the general population. There is also a significant increase in the number of yearly diagnosed cases of non-AIDS-defining cancers [1].

The incidence of these malignancies among PLWH remains elevated compared to that in uninfected population despite successful ART. Traditionally, this is linked to HIV-induced immune suppression with depletion of CD4+ T-helper cells, and exhaustion of lymphopoiesis, however, the immune suppression is much more complex than HIV-1 induced loss of CD4+ T cells. HIV-1 causes dysregulation of the innate immune system, persistent immune activation, dysfunction of the inflammatory response and immune system aging (senescence) early in HIV-1 infection. Successful ART ameliorates, but does not completely correct the major immune dysfunctions [2,3,4,5,6], substantial immunological impairment pertains even on the background of the successful ART [7,8] (for the latest review, see [9]). Hyper-immunoactivation and inflammation persisting in PLWH is recognized as a major cause of HIV-1 associated malignances. This abnormal immunoactivation emerges as the cumulative effect of thymic dysfunction, ART toxicity, persistent antigen stimulation caused by co-infections, microbial translocation, residual viremia and dysbiosis [10], aggravated by incomplete recovery of CD4+ T cell functions and intrinsic B and T cell defects on the background of persistent aberrant activation of monocytes, natural killer cells (NK) and innate lymphoid cells [7,11,12].The immune deficiency and dysfunction of the immune system may not be the only cause [13]. Under successful ART, HIV-1 should become latent, however, a study of HIV-1 integration sites in latently infected cell lines evidenced an ongoing viral replication [14], demonstrating that ART cannot fully suppress the process. Massive data have accumulated on the crucial role in high incidence of malignancies among PLWH of the residual virus production and circulating viral proteins. This review concentrates on their role in the high prevalence of cancers among individuals living with HIV-1.

## 2. Prevalence of Non-AIDS Defining Cancers Increases Despite Successful Antiretroviral Therapy

The category of non-AIDS-defining cancers (NADC) includes liver cancer related to infections with hepatitis B and C viruses (HBV and HCV), brain cancer, and cancers associated with infection with human papillomaviruses of high oncogenic risk (HR HPVs), specifically, the anal cancer.

### 2.1. Liver Cancer

Hepatocellular carcinoma (HCC) is the third-largest cause of cancer-related mortality on a global scale. It constitutes nearly the majority of liver cancer cases, followed by intrahepatic cholangiocellular carcinomas [15]. HCC is a recognized complication of liver cirrhosis, developing stepwise from regenerative to low-grade, then high-grade dysplastic nodules, although in some cases it may also develop de novo [15]. The burden of HCC is expected to increase worldwide in the next few decades, due to the population growth and aging expected in coming years [16]. Treated HIV-1 infection is associated with decreased survival in HCC, independent of stage, anticancer treatment, and geographical origins of the patients [16]. HIV-1 is not sufficient to cause liver cancer on its own, but may promote development of liver cancer by multiple mechanisms not yet fully understood [17]. Although the role of immune suppression in HCV-related HCC is not clear [17], mechanistic evidence suggests an accelerated progression of chronic liver disease to fibrosis and ultimately malignancy mediated by HIV-1-mediated impairment of antiviral CD4+ and CD8+ T-cell responses [18,19,20].

HIV-1 infection is characterized by increased microbial translocation resulting in elevated levels of circulating lipopolysaccharides (LPS) in the portal and systemic circulations [21]. LPS are well-known inducers of the innate immune activation. Increased levels of LPS and/or soluble CD14 (sCD14; reflects LPS-induced monocyte activation) in PLWH on ART correlate with impaired recovery of CD4+ T cells. They also tightly correlated with multiple markers of immune activation, specifically, high levels of type I interferons and activated CD8+ T cells. In HIV-1 infection, these two parameters strongly predict disease progression [21]. In the liver, LPS activate hepatic stellate (HSC) and Kupffer cells (KFC), resulting in the generation of superoxide and release of proinflammatory and profibrogenic cytokines such as TNF-α, IL-1, IL-6, and IL-12 that induce liver damage and accelerated liver fibrosis [22]. Activation of Kupffer cells by LPS involves signaling through TLR-4, shown to govern the transition from chronic hepatic inflammation to hepatocellular carcinoma [23]. Another product released from bacterial cell walls, (1→3)-β-d-Glucan (βDG), emerges as an additional significant source of monocyte and NK cell activation, further contributing to immune dysfunction and inflammation [24].

Growing evidence accumulates of HIV-1 grossly affecting the liver. HIV-1-monoinfected patients demonstrate markers of liver fibrinogenesis/liver injury (by transient liver elastography) correlated with high plasma levels of HIV-1 RNA [25]. HIV-1 RNA has been detected in primary human hepatocytes both ex vivo and in vitro [26,27]. Also many hepatocyte cell lines are permissive to a low level HIV-1 infection although the nature of receptor(s) for HIV-1 on liver cells is unclear [28]. HIV-1 can also directly infect Kupffer cells; infectious replication-competent HIV-1 has been isolated from KFC obtained from liver at autopsy from three HIV-1-infected individuals who died while on ART [29]. Another target of HIV-1 are hepatic stellate cells (HSC), the primary cells involved in liver fibrogenesis, affected through both direct HIV-1 infection and HIV-1 exposure [30]. Interactions of HIV-1, specifically its envelope protein gp120, with chemokine receptors CCR5 and CXCR4 induce cell signaling in HSCs and immune cells within the liver promoting inflammatory responses [31,32]. Direct HIV-1 infection of KFC results in the amplification of proinflammatory responses to LPS [33], enhanced fibrosis and cirrhosis, and exhaustion of virus-specific T-cells. HIV-1-infected HSCs produce collagen I and release monocyte chemoattractant protein-1 (MCP-1) [34]. Exposure of HSCs to HIV-1 results in the production of reactive oxygen species (ROS), and expression of collagen and tissue inhibitor of matrix metalloproteinases-1 (TIMP1) [35]. These events, together with abnormalities in the gut microbial communities, significantly contribute to the high rates of liver cancer in PLWH [36] (Figure 1).

Even more important driving force of hepatocellular carcinogenesis in PLWH is co-infection with HBV and HCV [37]. In HIV-1/HBV and HIV-1/HCV co-infected patients, HIV-1 infection decreases the rate of spontaneous viral clearance from the liver, accelerates fibrogenesis and increases the rates of liver-related morbidity and mortality, including the development of HCC [30,38]. In HIV-1/HCV co-infected individuals HCC occurs at a younger age and after a shorter period of HCV infection than in HIV-1 negative individuals [39], with the risk to develop HCC increasing each year by 11% [19]. Important risk factors for the progression to liver cancer are high HBV and HCV viral loads [18,19,40]. They are associated with (over)expression of viral oncoproteins known to induce oxidative stress and chromosomal instability/genomic damage, promote chronic inflammation with liver damage resulting in the malignant transformation of liver cells [41,42] (Figure 1).

### 2.2. Brain Cancer

PLWH are highly predisposed to developing brain cancer, including primary central nervous system lymphomas (PCNSL) and glioblastomas (GBM) [43,44]. In pre-ART era, brain tumors were registered in 10% of PLWH [43]. Prevalence of PCNSL in AIDS patients was 3600-fold greater than in the general population, reaching 12% in AIDS patients [44]. ART has dramatically reduced these rates, possibly due to the effect of protease inhibitors [45]. Still, the prevalence of brain tumors in PLWH appears to be higher than in general population: in USA; recorded prevalence of PCNSL in HIV-1 infected is 8.4% compared to <3.3% in the general US population [45,46] Also GBM occurs in PLWH (in various stages of HIV-1 infection) at a younger age and at a frequency 5.4- to 45-fold higher than in the general population [47]. Furthermore, the median survival rate in patients with GBM for PLWH is shorter than for HIV-1-negative patients receiving same treatment (an average of 8 compared to 14 months, respectively) [48].

The nature of the brain tumor-HIV-1 relationship is not fully understood. The majority of these tumors are central nervous system lymphomas but gliomas may develop as well. GBM tumors appear approximately three years after HIV-1 infection [43]. The stimulatory effect of HIV-1 infection on the development of GBM has been associated with reduced immune surveillance. However, survival of PLWH after GBM diagnosis is not associated with CD4+ cell counts [47]. The absence of a correlation between GBM development & progression with immune incompetence [47,49,50,51] indicates that aggressive tumor behavior is not a direct consequence of the immune deficiency and suggests direct involvement of HIV-1 in the initiation and progression of brain cancers. Importantly, HIV-1 infection in the brain is not limited to microglia/macrophages, but also affects astrocytes, which can then serve as a potential reservoir for further productive infection, viral persistence, and latency [52,53].

### 2.3. Squamous Cell Carcinomas

PLWH suffer from squamous carcinomas at numerous sites including the lung, anogenital region, oral cavity, epiglottis and cervix. Many of these malignancies are associated with infection by human papillomaviruses of high carcinogenic risk (HR HPVs). Similar to the rates of liver and brain cancer, the rates of HR HPV-associated cancers in PLWH are steadily growing despite successful ART [54,55,56]. CD4+ levels and resulting immune suppression play a prominent role in controlling HPV replication and development of early disease, particularly, the development of pre-cancerous intraepithelial neoplasia: in PLWH, the probabilities of HPV acquisition and development of intraepithelial neoplasia increase in proportion to the loss of CD4 T cells [57]. However, progression to high-grade lesions and further to cancer is not predetermined by CD4+ depletion, i.e., is not a straightforward outcome of HIV-1-induced immune suppression [58,59], but rather an outcome of the accumulated changes in the host cell genome and transcriptome involving tumor suppressor genes, apoptosis-related genes, DNA damage-repair genes, and cell cycle regulatory genes [58,60,61].

Question arises how could this rely to the epithelial cells which are considered to be non-susceptible to HIV-1 infection and non-permissive to HIV-1 replication? HIV-1 infects a variety of immune cells, such as CD4+ T lymphocytes and monocytes/macrophages. However, several studies show that it may also infect or rather “trespass” other cell types, in which HIV-1 virions and individual HIV-1 proteins were repeatedly detected. In primate models, application of HIV-1 to the surfaces of oropharyngeal [62], anal/rectal [63], cervicovaginal and foreskin/penile [62,64,65,66] epithelia was shown to lead to subsequent systemic infection of HIV-1-susceptible immune cells, indicating that HIV-1 travels through these tissues to reach its targets. Indeed, application of HIV-1 to human foreskin, vaginal and cervical tissue explants ex vivo leads to the transmission of HIV-1 across these epithelia [64,66,67,68,69,70,71].

These findings are not restricted to the epithelial cells of the reproductive tract. HIV-1 antigens and RNA were detected in gastric epithelial cells in the biopsy and autopsy samples of HIV-1-infected patients; furthermore, TEM analysis visualized HIV-1 particles in the cytoplasm of gastric epithelial cells [72]. Interestingly, HIV-1 load in blood positively correlated with the number of HIV-1-infected gastric epithelial cells. The latter increased with progression of chronic infection, being significantly higher at the AIDS compared to the asymptomatic stage. HIV-1 infection of gastric epithelial cells associated with a severe inflammatory response in the gastric mucosa manifested by infiltration and aberrant activation of the immune cells [72].

Another example is presented by human mammary epithelial cells (MEC). MEC express HIV-1 receptors CD4, CCR5, CXCR4, and galactosyl ceramide (GalCer). Although the evidence for direct MEC infection by HIV-1 was missing, HIV-1 virions were found in the endosomal compartments of these cells. Furthermore, activated CD4+ T cells co-cultured with HIV-1-exposed MEC were productively infected with HIV-1 [73]. This confirmed that mammary epithelial cells can endocytose HIV-1 and facilitate its transfer to CD4+ T lymphocytes [73]. At the other end, a contact-dependent HIV-1 transfer was shown from HIV-1-infected macrophages to both primary and immortalized renal tubule epithelial cells (RTE). Live imaging of HIV-1 infected RTE cells revealed four different fates: latency, hypertrophy, cell death, and proliferation [74]. HIV-1 can also enter airway epithelial cells and alter their function by increasing the expression of inflammatory mediators [75]. This data unequivocally demonstrate that HIV-1 could be internalized and/or sequestered by human epithelial cells of different origins.

## 3. Mechanisms Underlying HIV-1 Pathogenicity in Epithelial Cells

In CD4+ cells HIV-1 was reported to preferably integrate into cancer-associated genes or cell cycle regulation genes dysregulation of which can lead to cancer formation as was described for other retroviruses [76,77,78]. Replication of HIV-1 in epithelial cells has not been shown except for the early findings of human uterine epithelial cells productively infected by HIV-1 with reverse transcription of viral RNA, transcription of viral DNA, and secretion of infectious virus [79]. Of note, co-cultivation of human CD4+ T cell lines with HIV-1-infected uterine epithelial cells (and also by virions released by these cells) led to HIV-1 infection of the CD4+ T cells [79]. Bulk of data accumulated so far evidence sequestration of HIV-1 by human epithelial cells of different origins without evidence of productive replication or integration. However, a “real” infection can take place as well. HIV-1 was shown to hijack other viral Envs to directly enter CD4-negative cells through pseudotyping [80,81,82]. Lately, Tang Y. et al. have shown that HIV-1 infected T cells can fuse to and transfer the virus to placental trophoblasts, if the later express on their surfaces the envelope glycoprotein of human endogenous retrovirus family W1, syncytin [83]. This leads to the formation of an HIV-1 reservoir in the epithelial cells [83]. Syncytin-1 derives from a family of endogenous retroviruses and originates from HERVW1 infection of human germ cells [84]. Expression of syncytin could be a common feature of an epithelial cell which make them susceptible to HIV-1 via a “non-canonical” route of HIV-1 infection. These are not necessarily the epithelial cells of placenta. According to the recent preliminary report published in bioRxiv, HIV-1 can infect human bronchial epithelial cells; after exposition to HIV-1 they were shown to express p24 and contain latent HIV-1 provirus [85]. These findings along with the data by Asin SN et al. [79] indicate that in certain cases epithelial cells can be infected with HIV-1, possibly as a one-round abortive infection with reverse transcription of RNA and integration of the proviral DNA governed by respective enzymes constituting HIV-1 virion. Such integrated proviral HIV-1 DNA would not only serve as an HIV-1 reservoir, but would also give progeny to the genetically modified cells (with proviral DNA inserts) susceptible to malignant transformation. The observation by Hughes K et al. of a proliferation of HIV-1 infected epithelial cells consistent with clonal expansion of individual cells ideally fits this scenario [74].

HIV-1 antigens may also affect epithelial cells without infecting them. Epithelial cells may respond to the defective virions incapable of productive infection or freely circulating HIV-1 antigens shed by the infectious or defective virions. Addition of HIV-1/HIV-1 antigens to the epithelial cells generates an inflammatory microenvironment or rather microenvironmental immune abnormalities [86,87,88] (as those associated with HR HPV infection). Microenvironment of B-cell lymphomas in PLWH is characterized by expression of CD3, CD4, CD8, CD56, CD68, CD163, FOXP3, TIA1, granzyme B, perforin, CD57, CD34 and PD-1 [89], and enrichment with soluble factors, including cytokines IL-1, IL-2, IL6, IL10, and chemokines of the CCL and CXCL families [89,90]. Such microenvironment was also found in the intraepithelial cancerous lesions of PLWH [91]. Studies on the mucosa-associated lymphoid tissue system (MALT) in PLWH have shown abnormal immune responses in the mucosal milieu, including upregulation of expression of multiple regulatory cytokines such as IL-8, IL-23, TNF-α, IL-17A, and IFN-γ (TNF-α/IL-17A/IFN-γ triad), the depletion of Langerhans cells and CD3+ lymphocytes, increases in Foxp3+ T-regulatory cells, and in local lymphocyte infiltrates composed by CD8+ T cells, associated with the development of high-grade squamous intraepithelial lesions (HSILs) [91,92,93,94].

Furthermore, presence in the epithelial cells of HIV-1/HIV-1 proteins modulates their capacity to express E-cadherin, a marker of epithelial to mesenchymal transition (EMT) [69,75,95]. HIV-1 interaction with the surface of mucosal epithelial cells was also shown to activate the transforming growth factor-beta (TGF-β) and mitogen-activated protein kinase signaling pathways [96]. When activated, these pathways may lead to the disruption of epithelial junctions and EMT [97]. Indeed, EMT was induced by exposure of oral keratinocytes from HIV-1-negative individuals to HIV-1 virions as well as Tat and gp120 proteins [98]. Within premalignant cells or in the environment of the malignant cells, HIV-1 driven EMT would promote motile/migratory cells and accelerate the neoplastic process.

Altogether these observations imply direct carcinogenic effect(s) of HIV-1 virions and/or antigens. This concept, proposed in 2002 by B Clarke & R Chetty [58] and four years later by Palefsky JM [59], is now supported by considerable experimental proof. It brings up several issues of importance for epithelial cells: (i) could malignant transformation be promoted by cooperation of HIV-1 with other oncogenic viruses; (ii) which HIV-1 antigens are implicated; and finally (iii) what are the underlying molecular mechanisms?

## 4. Potentiation of Carcinogenesis by Interactions of HIV-1 with Other Oncogenic Viruses

Oncogenic transformation associated with virus infection was for a long time considered to result from a mono-infection (infection with a single virus). However, it is now established that in many cases induction of cancer depends on the simultaneous presence and interactions of multiple viral agents in diverse combinations. Viruses co-infecting human tissues may have synergistic or regulatory effects on carcinogenesis, targeting existing neoplastic cells as well as their microenvironment including reactive T-cells, B cells and macrophages, and non-immune cells such as endothelial cells. HIV-1, in particular, potentiates the effects of EBV, KSHV, HCV, and HPV oncogenes, promoting carcinogenesis in individuals co-infected with HIV-1 and EBV, KSHV, HCV, HBV, and HPV. Here, we will focus on molecular interactions of HIV-1 with HBV, HCV, and HPVs.

Progression to liver cancer/HCC in HIV-1/HBV and HIV-1/HCV co-infected patients is promoted by direct and indirect interactions between these viruses and their antigens within the cells harboring HIV-1 due to infection or sequestration of the virion (viral proteins). HIV-1 infection of hepatic cell lines increases the expression of HBV antigens [27]. HIV-1 gp120 causes intracellular accumulation of HBV DNA as well as HBsAg causing hepatotoxicity [99]. Direct interaction of HIV-1 and HBV in liver cells has been demonstrated, with the HBV X protein interacting with HIV-1 Tat to facilitate HIV-1 replication [99]. Upon co-cultivation of HIV-1 infected Jurkat cells with hepatocytes, up-to 16% of the latter acquire Nef. Sequestered Nef alters the size and numbers of lipid droplets (LD), inducing 1.5 to 2.5 fold up-regulation of replication of HCV subgenomic replicon, a remarkable finding in relation to the initially indolent viral replication. Nef also dramatically enhances the ethanol-mediated up-regulation of HCV replication accelerating progression to HCC [100]. HIV-1 gp 120 also causes TGF-β mediated up-regulation of HCV replication [86]. Taken together, these data indicate that HIV-1 and single HIV-1 proteins are critical elements in accelerating progression of liver pathogenesis by enhancing HBV and HCV replication and coordinating production of key intra- and extra-cellular molecules that orchestrate liver decay [100].

One of the mechanisms of HIV-1 potentiation of liver cancer is the induction of oxidative stress. HCV, HIV-1 (and antiretroviral therapy) act together to activate production of ROS in HSCs and hepatocytes. ROS promote phosphorylation of the major mitogen-activated protein kinases active in human cells, p38 kinase, c-JUN N-terminal kinase (JNK) and extracellular signal-regulated kinase (ERK) that control cell growth, differentiation and apoptosis. In their turn, the phosphorylated p38 MAPK, JNK, and p42/44 ERK phosphorylate nuclear factor kappa-light-chain-enhancer of activated B cells (NF-κB) protein complex, mastering transcriptional regulation of inflammation and cell death [31]. Following these events, phosphorylated NF-κB translocates to the nucleus, and where it normally modulates the production of both pro- and antifibrogenic/antiapoptotic genes, ensuring that liver cells are protected from apoptosis, but are capable to build the required inflammatory and immune responses [101]. In the presence of LPS, NF-κB can upregulate the expression of profibrogenic genes, such as procollagen α1, transforming growth factor β1 (TGF-β1) and tissue inhibitor of MMPs (TIMP-1) [31,101]. This process is accelerated by HIV-1/HIV-1 proteins: exposure of hepatocytes to HIV-1/HIV-1 proteins results in the elevated production of ROS and increased expression of collagen and TIMP1, further amplified by HCV infection, and even exposure to infectious HCV [35]. Taken together, these data indicate that HIV-1-mediated potentiation of hepatocellular carcinogenesis reflects a concerted action of HIV-1, HBV and HCV as viruses and/or individual viral proteins (Figure 1). Based on compelling data, McGivern & Lemon even suggested that the path to hepatocellular carcinoma in chronic hepatitis C shares important features with the carcinogenesis induced by HPV [102].

The increased risk of PLWH developing HPV-associated cancer can also, at least in part, be due to the interactions between HIV-1 and HPV. In general, epithelial cells of PLWH show loss of E-cadherin, and upregulation of vimentin and TGF-b1 expression with spindle-like morphology indicating induction of TGF-b1-dependent EMT, critical for malignant transformation. As noted above, EMT is induced not only due to HIV-1 infection, but also through exposition of epithelial cells to HIV-1 proteins [69,75,95,97]. EMT-induced keratinocytes can then be infected with pseudoviral HPV16 particles (HPV-16 PsVs) and whole HPV16 virus, with infected cells expressing viral oncogenes E6/E7, whereas unexposed keratinocytes could not be infected with either PsVs, or infectious HPV16. Furthermore, “HIV-1-induced” EMT keratinocytes could be transformed with HPV16 DNA, transformed cells showing active proliferation and migration [103]. This confirms that prolonged exposure to and interaction of HIV-1 with oral and anal epithelial cells induces EMT. EMT-induced loss of cell adhesion and increased proliferation and mobility of epithelial cells play a critical role in HPV infection and HPV-associated transformation. HIV-1-induced EMT in the orogenital mucosa may promote progression of pre-cancerous HPV-associated neoplasia to cancer in HIV-1-infected individuals [103].

“Molecular” cooperation between HIV-1 and HPV has not been sufficiently well characterized, but there are relevant examples in this field. Tat protein was shown to transactivate the HPV long control region and increase expression of oncoprotein E7 of HPV18 in HeLa cells [104,105]. Tat can upregulate the expression of E6 and E7 oncoproteins of HPV type 16 in HPV 16-infected human oral keratinocytes, notably enhancing the in vitro proliferative capacity of these cells [106,107], and increase the transcription of E2 modulating HPV replication [108]. The direct angiogenic effects of Tat [109] or its capacity to up-regulate the expression of E6 and E7 of HR-HPVs [110] allows Tat to favor the angiogenic switch in high-grade CIN. We have shown that gp120 and reverse transcriptases (RT) derived from various HIV-1 strains, can increase the expression of HPV 16 E6 in a cervical cancer cell line containing full-length HPV 16 genome Ca Ski (Figure 2), while HIV-1 p24 exerts no effect. In similar conditions, gp120 increases the expression of HPV16 E6 also in HPV16 immortalized anal epithelial AKC2 cells [104,106,111,112]. Furthermore, Tugizov et al. have shown that in the HPV-immortalized anal and cervical epithelial cells Tat and gp120 proteins induce the EMT phenotype, leading to increased migration of cells via collagen membranes [103]. The data on the interaction(s) between HPV and other HIV-1 proteins is missing.

Overall, these findings indicate that the increased incidence of AIDS-defining and non-AIDS defining forms of cancer in PLWH may reflect the direct or indirect, often concerted, carcinogenic effect(s) of HIV-1 and/or individual HIV-1 proteins on diverse infected as well as uninfected bystander cells. Furthermore, some HIV-1 proteins appear to be directly involved in cell transformation and propagation of malignant cells.

## 5. HIV-1 Antigens Involved in Cell Transformation and Tumor Propagation

### 5.1. Transactivator of Transcription (Tat)

Tat has long since been known to influence cell cycle progression. In HeLa cells, Tat induces a significant increase in the levels of proliferation markers together with the reduction in the expression of cell cycle inhibitors of transcription [119]; it inhibits epithelial differentiation, blocks apoptosis in vitro and accelerates tumor formation in vivo [119]. In addition, Tat significantly increases in vitro migration in the absence of fetal calf serum [119]. These results suggest that HIV-1 may enhance carcinogenesis by promoting cell cycle progression [111]. Furthermore, it has been shown that binding of Tat to Tat-interacting promoter 30 (TIP30) enhanced EMT and metastasis of non-small cells lung cancer cells by regulating the nuclear translocation of Snail [120]. One of the possible mechanisms of Tat induced carcinogenesis is blocking at the mRNA level of the expression of a Rb family member pRb2/130 and cyclin-dependent kinase inhibitors p21 and p17 [111]. The transduction domain of Tat specifically attenuates growth of polyamine-deprived tumor cells [121]. Tat is also known to modulate VEGF and targets VEGFRs which increases angiogenesis and supports tumor growth [122]. Furthermore, Tat alters DNA repair in host cells, potentially leading to genomic instability [123,124]. Specifically, Tat induces expression of the DNA polymerase beta gene, which codes for a central mediator in the DNA base-excision repair pathway [125]. It also interferes with double-strand break DNA repair, as cellular extracts containing Tat possess a reduced capacity to re-join linearized DNA [126], indicating that Tat, as well as cellular co-factors of Tat, interfere with repair of double-stranded DNA breaks [123].

### 5.2. Envelope Glycoprotein gp120

Glioma cells were shown to interact with the HIV-1 envelope protein gp120. This interaction promotes proliferation, migration, survival and stimulates glycolysis in glioma cell lines and tumor growth in animal models [127]. Increased glycolysis, also known as the Warburg effect characteristic of malignancy [128], results in increased protein and lipid synthesis, and promotes uncontrolled propagation (both proliferation and invasion) of tumor cells, as it provides them with glycolytic intermediary precursors required for the synthesis of DNA, proteins and lipids [127,129]. As Tat, gp120 induces EMT and cell migration through the TGF-B1 and MAPK signaling pathways [115,130].

### 5.3. Accessory Protein Negative Factor (Nef)

Nef is one of the earliest and most abundantly expressed HIV-1 proteins. Nef has the ability to modulate multiple cellular signaling pathways in both CD4+ lymphocytes and macrophages. Nef inhibits the apoptotic function of p53 due to its ability to decrease p53 protein half-life and, consequently, p53 DNA binding activity and transcriptional activation [131]. Both internalized and ectopic expression of Nef in endothelial cells synergizes with Kaposi’s sarcoma (KS) KSHV oncoprotein K1 to facilitate vascular tube formation and cell proliferation, and enhance angiogenesis in the chicken chorioallantoic membrane (CAM) model. In vivo experiments further indicate that Nef can accelerate K1-induced angiogenesis and tumorigenesis in athymic nu/nu mice [132]. On non-small lung cancer A549 cells, Nef promotes cell proliferation, migration, anchor independent growth and reduces the levels of expression of p53, increasing the aggressiveness of cancer cells [133].

### 5.4. Reverse Transcriptase (RT)

We have shown that constitutive expression of HIV-1 RT in murine mammary gland adenocarcinoma 4T1 cells leads to upregulation, in a concentration-dependent manner, of the expression of the transcription factors Twist and Snail tightly involved in EMT [134]. In vivo, expression of RT by 4T1 cells results in enhanced tumor growth and potentiates formation of metastasis in distal organs of immunocompetent syngenic mice [134]. Interestingly, this is not a common property of the reverse transcriptases, as constitutive expression of enzymatically active reverse transcriptase domain of telomerase reverse transcriptase, on contrary, suppressed both tumor growth and metastatic activity of 4T1 cells [116].

### 5.5. Matrix Protein p17

Matrix/p17 protein induces expression of chemokines [135], exerts pro-angiogenic [136] and lymphangiogenic [137] activities, and deregulates the biological activity of diverse cells of the immune system [138]. Overall, p17 generates a prolymphangiogenic microenvironment, predisposes the lymph node to lymphoma growth and metastasis [137] and promotes the aggressiveness (propagation) of human triple-negative breast cancer cells [139]. In a HIV-1 transgenic mouse model of lymphoma, only expression of HIV-1 p17, but not of other HIV-1 proteins, induced spontaneous B-cell lymphomas in HIV-1 transgenic mice, with p17 expressed at high levels in the early stages of the disease [140]. Murine lymphoma tissues exhibited enrichment in expression of the recombination-activating genes (Rag1/2) [140]. The latter suggests that intracellular signaling induced by p17 leads to genomic instability and promotes the transformation [140].

Thus, several HIV-1 proteins are directly or indirectly oncogenic, stimulating transformation of healthy cells and propagation and aggressiveness of already existing cancer cells. These oncogenic properties are linked to two essential characteristics of these proteins: their capacity to induce oxidative stress with production of reactive oxygen species and their ability to exit HIV-1-infected cells (active or passive transport).

## 6. Oncogenic HIV-1 Proteins Induce Oxidative Stress

Virally-induced cancer evolves over long periods of time in the context of a strongly oxidative microenvironment, on the background of chronic inflammation. Oxidative stress induced by chronic viral infection is one of the factors driving neoplastic transformation, ultimately leading to oncogenic mutations in many cellular signaling cascades that drive cell growth and proliferation [42,141]. Oxidative damage of chromosomal DNA and chronic immune-mediated inflammation are key features of HBV, HCV, HPV, and HIV-1 infections [42,141]. As we have earlier reviewed, numerous lines of evidence show that HIV-1 infection triggers pronounced oxidative stress in both laboratory models and the context of in vivo infection by deregulation of oxidative stress pathways with escalation of ROS production and by inducing mitochondrial dysfunction [141]. As a result, PLWH exhibit multiple markers of oxidative stress including DNA damage [134,142]. The enhancement of ROS production is mediated by the envelope protein Gp120, Tat, Nef, RT, and p17 [141,142,143,144,145,146].

### 6.1. Transactivator of Transcription 

Tat induces oxidative stress both directly and indirectly via several independent mechanisms. The first involves the NADPH oxidases [147], and in the second, an enzyme involved in the catabolism of biogenic polyamines, spermine oxidase (SMO) [148], and the third, a mitochondrial dysfunction [149]. A detailed analysis of the levels of ROS in different subcellular compartments of the HIV-1 infected cells revealed a strong increase in the levels of H_2_O_2_ in the endoplasmic reticulum (ER), demonstrating with the help of genetically encoded ratiometric sensor HyPER [150,151]. This indicated the involvement in H_2_O_2_ production of NOX4 which primarily resides in ER [152]. The levels of H_2_O_2_ in the cytoplasm and mitochondria were not elevated [151]. The above activities of Tat are thought to underlie the onset of HIV-1-associated dementia [109,150].

### 6.2. Envelope Protein Gp120

Early findings indicated that gp120 increases free radical production from monocyte-derived macrophages (MDM) detected by spin-trapping methods, and that the spin trap adduct results from a reaction involving nitrogen oxide NO or its closely related oxidized derivatives [153]. We have earlier summarized a profound role of gp120 in the induction of oxidative stress [141], namely gp120 induces ROS production in cell lines of lymphoid origin, in the endothelial brain cells, astrocytes, neurons and microglia. In astrocytes, it enhances ROS production by several parallel mechanisms: via Fenton–Weiss–Haber reaction, NOX2 and NOX4, and cytochrome P450 2E1 (CYP2E1) [154,155]. The latter is mediated through the upregulation of CYP2E1 expression. In cancer (neuroblastoma) cells, gp120 induces proline oxidase that synthesized pyroline-5-carboxylate with concomitant generation of ROS (reviewed in [141]).

The effect of HIV-1/HIV-1 proteins on the cellular antioxidant defense system is controversial. They can both suppress and enhance antioxidant defense pathways [141]. Gp120 was shown to induce oxidative stress response. It up-regulates functional expression in cultured astrocytes of multidrug resistance protein 1 (Mrp1) which effluxes endogenous substrates glutathione and glutathione disulphide involved in cellular defense against oxidative stress [156]. It also upregulates the expression of nuclear factor erythroid derived 2-related factor 2 (Nrf2), a basic leucine zipper transcription factor which is known to regulate antioxidant defensive mechanisms) in human astrocytes, stimulating expression of key antioxidant defensive enzymes hemoxygenase (HO-1) and NAD(P)H dehydrogenase quinone1 (Nqo1) [157]. Pre-treatment of astrocytes with antioxidants or a specific calcium chelator BAPTA-AM, significantly blocks the upregulation of Nrf2, HO-1 and Nqo1 [157].

### 6.3. Accessory Protein Negative Factor

Nef protein has pro-oxidant activity in microglial cells and in neutrophils. It first induces phosphorylation and then translocation of the cytosolic subunit of NADPH oxidase complex p47(phox) into the plasma membrane which in turn induces superoxide anion release from macrophages [158,159]. As a multifunctional HIV-1 protein, Nef also activates the Vav/Rac/p21-activated kinase (PAK) signaling pathway involved in activation of phagocyte NADPH oxidase (thus, Nef indirectly activates NADPH oxidase) [160]. This leads to the dramatic augmentation of the production of ROS [100], and enhancement of cell responses to a variety of stimuli (Ca(2+) ionophore, formyl peptide, endotoxin) [160]. It also leads to decreased tolerance of the cells to hydrogen peroxide, specifically in astrocytes which normally support neuronal function and protects them against cytotoxic substances including ROS [161]. Rac1-dependent NOX2-mediated reactive oxygen species production was shown to contribute to ongoing HIV-1-related vascular dysfunction [162].

### 6.4. Reverse Transcriptase

We have previously demonstrated that expression of RT by human cells induces production of ROS [163]. Later studies demonstrated that this is a property of different RT variants, including drug resistant variants, and variants retargeted for lysosomal processing and secretion [114,163]. Expression of all RT variants led to an increase in the levels of expression of Phase II detoxifying enzymes HO-1 and Nqo-1. Artificial secretion of RT resulted in a decrease of RT capacity to induce oxidative stress with a decrease in the production of ROS compared to the parental enzyme [114].

### 6.5. Matrix Protein p17

There is no direct evidence of p17-induced oxidative stress. However, p17 possesses specific structural motifs defined as “*coiled coil*” sequences, and has a high propensity to form multimers, mis-fold and aggregate, forming amyloidogenic assemblies [164,165]. This is typical to amyloidogenic proteins actively involved in the pathogenesis of many human diseases, such as Alzheimer’s disease and Parkinson’ disease. Amyloidogenic assemblies are toxic, specifically to neural cells. Experiments in the invertebrate nematode *Caenorhabditis elegans* as a “biosensor” demonstrated that p17 significantly inhibits its pharyngeal contractions as do the amyloidogenic proteins [166]. Intrahippocampally injected into mice, p17 induced neurocognitive disorders, comparable in strenght to the effects of other known amyloidogenic proteins [166]. Interestingly, amyloidogenic proteins (typically amyloid-beta peptide Aβ) bound to redox active metal ions, such as copper, catalyse the production of ROS, in particular the most reactive one, hydroxyl radical. This effect may underlie the observed oxidative damage exerted by Aβ peptide on itself and on the surrounding molecules (proteins, lipids, DNA) [167]. One can hypothesize that matrix protein p17 with its amyloidogenic assemblies may trigger the production of ROS through a similar mechanism.

Thus, HIV-1 proteins with known oncogenic/mitogenic potential, Tat, gp120, Nef, RT, and potentially p17, have a potential to directly or indirectly induce oxidative stress, which could be one of the mechanisms by which they induce and potentiate carcinogenesis (Figure 3). Interestingly, HIV-1 proteins with an oncogenic potential involved in the induction of oxidative stress, such as Tat, gp120, Nef, RT, and possibly p17, can be found outside of the cells in which they are expressed.

## 7. Oncogenic HIV-1 Proteins Inducing Oxidative Stress Are Found in the Extracellular Space

### 7.1. Transactivator of Transcription 

Tat protein can be produced and released into the extracellular space by cells harboring actively replicating HIV-1 as well as by latently infected cells, with further uptake by the neighboring uninfected cells. Uptake of Tat would result in upregulation of inflammatory genes and cytotoxicity; this scenario was observed in a number of HIV-1 associated comorbidities, specifically, in neurocognitive disorders, cardiovascular impairment and accelerated aging [169]. Dangerously, the process may occur on the background of successful ART, in the absence of active HIV-1 replication and viral production. Considering that approximately 2/3 of all Tat expressed by infected T cells is secreted [170], the activities of Tat described above make a considerable contribution into HIV-1 associated pathologies [171,172].

Soluble Tat, in the absence of the virus, has been shown to cause induction of apoptosis, release of neurotransmitters, oxidative stress and inflammation [169]. Uptake of Tat has been shown to lead to activation of several transcription factors [173,174] including Sp1, NF-κB, and others, resulting in the modulation of expression of both HIV-1 and host genes, including pro-inflammatory cytokines (like TNF-α, CCL2, IL-2, IL-6, and IL-8), adhesion molecules and sometimes, and pro- and anti-apoptotic factors [175,176,177,178,179], p53 and HPV oncoprotein E6 [107].

### 7.2. Envelope Protein gp120

Envelope protein gp120 is known to be secreted by chronically infected cells [180,181], particularly from the intraepithelial immune cells even in presence of ART [98]. A subset of PLWH demonstrate persistent circulation in plasma of gp120 [182] and in saliva [98]. Moreover, gp120 was found in tissues of PLWH [183]. Brain cells can be directly exposed to gp120 secreted by infiltrated and infected microglia and astrocytes [127]. Gp120 is internalized by bystander cells through receptor-independent mechanisms [184]. Internalization of gp120 leads to the release of several proinflammatory, angiogenic, and lymphangiogenic factors from affected cells [185].

### 7.3. Accessory Protein Negative Factor 

Accessory protein negative factor Nef is found in the serum of PLWH [186,187]. Nef can stimulate its own export via the release of extracellular vesicles (exosomes) from HIV-1 infected cells [188]. Of note, exosomes serve as a marker and confirmation of the systemic oxidative stress [189]. Secreted in exosomes, Nef triggers apoptosis in bystander cells. Extracellular Nef has deleterious effects on CD4+ T cells [188,190]; on bystander B cells by suppressing immunoglobulin class switching [191]; and on astrocytes [192] and endothelial cells [162].

### 7.4. Reverse Transcriptase (RT) 

In our lab, we have shown secretion of RT into cell culture fluids of cells transiently expressing RT [114]. Recently, RT was also detected in the exosomes detected in the uterine of PLWH [193].

### 7.5. Matrix Protein p17

Matrix protein p17 is continuously released into the extracellular space from HIV-1-infected cells, and can be detected in the plasma of PLWH and in different organs and tissue specimens [138]. Cellular aspartyl proteases promote the unconventional secretion of biologically active p17 [194]. HIV-1 secretion of biologically active p17 takes place at the plasma membrane and occurs following its interaction with phosphatidylinositol-(4,5)-bisphosphate and its subsequent cleavage from precursor Gag (Pr55Gag) by cellular aspartyl proteases [194]. Extracellularly, p17 deregulates the function of different cells involved in AIDS pathogenesis. Importantly, p17 accumulates and persists in different organs and tissues of PLWH on ART, even in the absence of any replicative activity [136,195,196]. These findings strongly suggest that p17 may be chronically present in HIV-1-I infected cells and tissues, even under ART-associated suppression of HIV-1 replication.

Thus, gp120 and Tat are actively secreted into the endothelial cell micro-environment, Nef can be neighboring uninfected cells including cells which cannot be infected with HIV-1, modulating their metabolism, cell cycle progression, ability to differentiate, motility, and, importantly, the genomic stability, through induction of ROS. Some HIV-1 proteins such as matrix p17 and gp120 can accumulate and persist in lymphoid tissues for at least 1 year after the on-start of ART on the background of successful suppression of viral replication [196]. These proteins are involved in different processes associated with malignant transformation and tumor growth with significant direct and indirect adverse effects on the epithelial cells. These include a range of responses that contribute to endothelial dysfunction, including enhanced adhesiveness, permeability, cell proliferation, apoptosis, as well as activation of cytokine secretion [86], eventually leading to malignant transformation (Figure 3). In this respect, their effect would resemble oncogenesis mediated by known viral oncoproteins originating from EBV, HTLV-1, KSHV, HCV, HBV, HPV, and identified as causative agents of both AIDS-defining and non-AIDS defining forms of cancer.

## 8. Conclusions

People living with human immunodeficiency virus receiving antiretroviral therapy are characterized by high prevalence of different forms of cancer affecting epithelial cells. HIV-1 does not infect epithelial cells, however both HIV virions and proteins were shown to be sequestered into epithelial cells and affect their functions. These proteins have three specific properties:First, HIV proteins Tat, Nef, gp120, matrix protein p17, reverse transcriptase/RT induce oxidative stress with serious consequences in the form of DNA, protein and lipid damage, as well as changes in the intracellular signaling.Second, Tat, Nef, gp120, matrix protein p17, RT have a direct carcinogenic potential as demonstrated in the series of in vitro experiments and experiments in the laboratory animals.Third, Tat, Nef, gp120, matrix protein p17, reverse transcriptase/RT were shown to exit HIV expressing cells by different mechanisms, and, once present in the extracellular space, can be up-taken by innocent neighbor cells.

Sequestered/internalized by innocent bystander cells, these proteins modulate their metabolism, cell cycle progression, ability to differentiate, motility, redox balance (induce ROS) and genomic stability. Through this, they can trigger malignant transformation of normal cells. Another outcome is propagation (proliferation and dissemination) of already existing precancerous and cancer cells, and enhanced growth and metastatic activity of tumors expressing or exposed to HIV-1 proteins.

Altogether, we present a new mechanism of HIV-associated malignant transformation of epithelial cells driven by individual HIV proteins through the induction of reactive oxygen species. In this scenario, HIV-1 proteins act in a manner similar to the known viral oncogenes, and can cooperate with them promoting KSHV, EBV, HBV, HCV, and HPV-associated carcinogenesis. Such pathway of HIV associated carcinogenesis can co-occur together with carcinogenesis driven by persistent immune inflammation, and dysfunction of B cells, T cells and cellular components of the innate immune system.

## Figures and Tables

**Figure 1 cancers-13-00305-f001:**
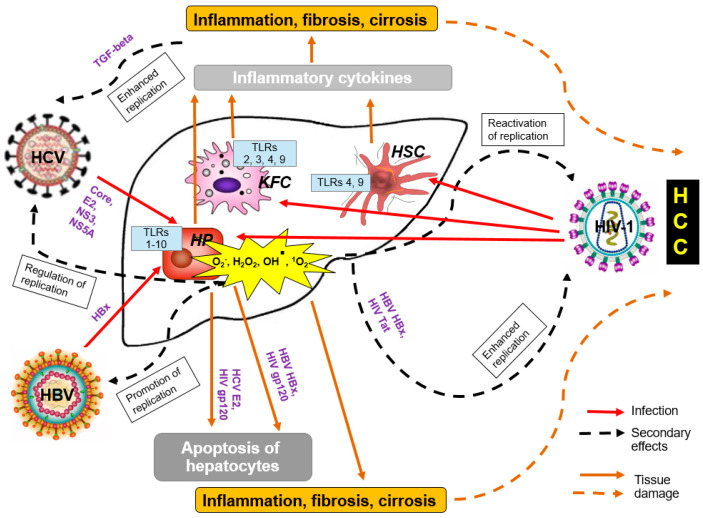
The effect of HIV-1 on cells of the liver. Infection with HIV-1 and even exposition of hepatocytes (HP), hepatic stellate cells (HSC), Kupffer cells (KFC) to HIV-1 leads to production of reactive oxygen species (ROS) and induction of proinflammatory microenvironment, which in turn, promote/enhance replication of HBV, HCV, as well as HIV-1 itself, resulting in enhanced fibrosis, cirrhosis and development of hepatocellular carcinoma (HCC). Infections are depicted in red, secondary effects in dashed black, and events leading to tissue damage in ochre-colored lines.

**Figure 2 cancers-13-00305-f002:**
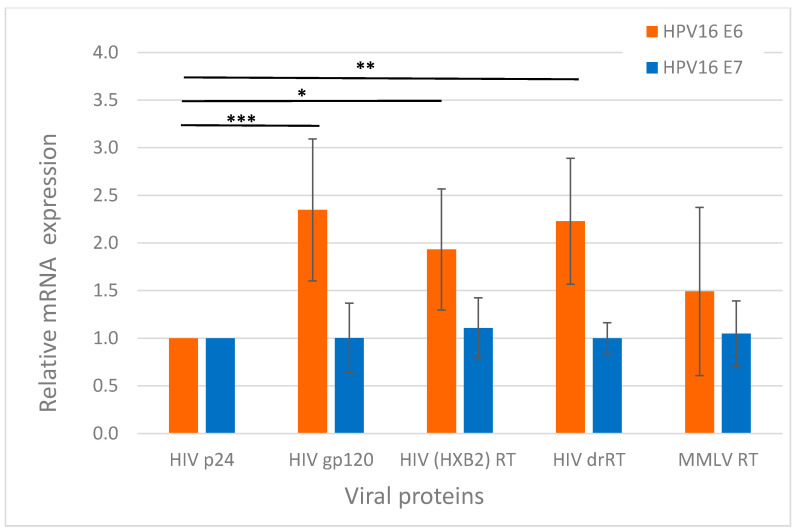
Transcription of oncoproteins E6 and E7 of HPV 16 in Ca Ski cells treated with HIV-1 proteins. Ca Ski cells harboring 600 full genomic copies of HPV 16 (ATCC CRL-1550) were cultured in RPMI-1640 medium (PanEco, Moscow, Russia) supplemented with 10% FBS and 100 mg/mL penicillin/streptomycin mix at 37 °C in an 5% CO_2_ and split every 4 days. A panel of recombinant HIV-1 proteins: gp120 [113]; p24 (NIBSC ARP 694.1); RT of HIV-1-1 clade B HXB2 strain [114], drug resistant (dr) RT of HIV-1-1 clade B isolated from patient with multiple drug resistance mutation (RT1.14; [114]) and RT of Moloney murine leukemia virus (MMTV) (CRIE, Moscow, Russia) were added to the culture medium, typically in concentration of 1 ng/mL, and incubated for 48 h, according to the methodology described previously by Lein K. et al. [115] Total RNA was extracted and reverse transcribed as described by Jansons et al. 2020 [116]. Gene-specific PCRs were performed on Rotor-Gene 6000 (Qiagen, Darmstadt, Germany) with SYBR Green kit (Evrogen, Moscow, Russia) with primers specific to HPV 16 E6 and E7 [117]. Expression of mRNA, assessed by the standard ddCt method, was normalized to expression of 18S RNA (18Srna_rt_f: GTAACCCGTTGAACCCCATT; 18Srna_rt_r: CCATCCAATCGGTAGTAGCG), and presented as fold change compared to cells treated with p24, as was recommended earlier [118]. Values represent mean ± SD from two independent assays run in duplicates. *** *p* < 0.001, ** *p* < 0.01, * *p* < 0.05 by the ordinary two-way ANOVA with Sidak’s multiple comparisons test.

**Figure 3 cancers-13-00305-f003:**
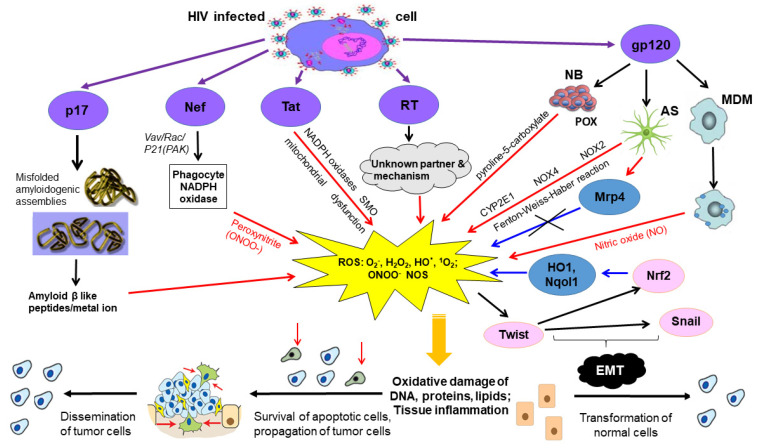
Suggestive mechanism of direct carcinogenic effects of HIV-1 proteins. HIV-1 infected cells express and release gp120, Tat, Nef, p17, RT, each capable of the induction of oxidative stress. (1) p17 may trigger the production of ROS through binding of redox active metal ions by its amyloidogenic assemblies [167]. (2) Nef may indirectly activate NADPH oxidase by activating the Vav/Rac/p21-activated kinase (PAK) signaling pathway involved in phagocytic NADPH oxidase activation and produce peroxynitrite [160]. (3) Tat induces oxidative action through several independent mechanisms via NADPH oxidase, spermine oxidase (SMO) induction and mitochondrial dysfunction [148]. (4) RT induces ROS through unknown mechanisms. There is ROS –dependent activation of the Twist [134], which regulates the expression of Nrf2, which stimulating the expression of antioxidant enzymes (HO1, Nqol1). In addition, the Twist regulates the expression of the Snail. Both transcription factors, Twist and Snail, are involved in epithelial to mesenchymal transduction (EMT). (5) Gp120 increases free radical production from monocyte-derived macrophages (MDM) inducing nitrogen oxide (NO). In astrocytes (AS), it enhances ROS production by several parallel mechanisms: via cytochrome P450 2E1 (CYP2E1), NOX2 and NOX4, and the Fenton-Weiss-Haber reaction. Multidrug resistance proteins (Mrps) involved in cellular defense against oxidative stress. Mrp4 (isoform of Mrp) involved in the regulation of ROS and it acts against ROS [156]. In neuroblastoma cells (NB) gp120 was shown to induce proline oxidase that produces pyroline-5-carboxylate with a concomitant generation of ROS [141]. Production of ROS, which damage of bystander cells inducing oxidative damage of DNA, proteins and lipids, apoptosis and inflammation. DNA damage drives genomic instability and promotes transformation of healthy cells, and propagation and dissemination of malignant cells [168]. Arrows indicate: purple arrows—secretion/entering the intercellular space; black arrows—relationships and interactions; red arrows—production of ROS; blue arrows—oxidative stress response. Text above arrows designates the processes leading to the production of ROS, and text below the arrows, forms of ROS.

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
