# Peer review of "Oncogenic Effects of HIV-1 Proteins, Mechanisms Behind"

_cancers, 2021, doi:10.3390/cancers13020305_

Round 1
Reviewer 1 Report
This review presents the case that HIV-associated carcinogenesis results from direct cellular effects of HIV and HIV proteins. This is one of the possible explanations; others include immune defects allow oncogenic virus to establish stable infections, the ongoing immune/inflammatory response involved in a fighting a chronic, systemic viral infection creates fertile ground for carcinogenesis, or some combination of the above. The mechanism presented in this manuscript is distinctly a minority view, but well deserving of a hearing.
The authors have thoroughly marshalled their arguments and reviewed the literature. The have made their case in a well-organized fashion. Several different plausible mechanisms whereby HIV or its products may directly cause cancer are explored.
My only real critique is that while the authors do give voice to the other explanations listed above, they summarize after reviewing all the evidence associating HIV with malignancies the following: “Altogether these observations imply direct carcinogenic effect(s) of HIV-1 virions and/or antigens. This concept, proposed in 2002 by B Clarke & R Chetty [56] and four years later by Palefsky JM [57], is now supported by considerable experimental proof” (lines 230-232). In fact, there are a number of better accepted explanations. The authors need to make it explicitly clear that the explanation expounded in this review is only one of several potential explanations.
Author Response
Authors wish to thank Reviewer 1 for objective evaluation of our review. We completely agree that the mechanism presented in this manuscript represents a minority of the cases of cancer associated with HIV-1. Indeed, HIV-associated cancers affect multiple organs and tissues (brain, liver, lungs, hematopoietic system etc.) involving cells of different origins. Epithelial cells constitute just one of the populations.
The main accepted mechanism of HIV associated carcinogenesis is hyper-immunoactivation and inflammation. This is the cumulative effect of thymic dysfunction, ART toxicity, persistent antigen stimulation caused by co-infections, microbial translocation, residual viremia and dysbiosis. (Zicari S, Sessa L, Cotugno N, Ruggiero A, Morrocchi E, Concato C, Rocca S, Zangari P, Manno EC, Palma P. Immune Activation, Inflammation, and Non-AIDS Co-Morbidities in HIV-Infected Patients under Long-Term ART. Viruses. 2019 Feb 27;11(3):200. doi: 10.3390/v11030200. PMID: 30818749; PMCID: PMC6466530), aggravated by incomplete recovery of CD4+ T cell functions and intrinsic B and T cell defects on the background of persistent aberrant activation of monocytes, natural killer cells (NK) and innate lymphoid cells. The description of this major pathway of carcinogenesis in ART treated PLWH has been added to the introduction and mentioned in the summary. The mechanism of the carcinogenesis described in the review associated with the direct oncogenic effects of HIV proteins inducing oxidative stress is not an alternative, but a complementary possibility. We hoped that the revisions we made in the text made this explicitly clear.
Reviewer 2 Report
The authors reviewed the oncogenic effects of HIV proteins and mechanisms of action while including one figure of their own data without providing any methodology. The article is supported by nearly two hundred references. In this reviewer’s opinion, the article could benefit from restructuring, reducing its size and repetitiveness and somehow crystallising or conveying more clearly and concisely its main messages.
Author Response
We thank the Reviewer for the valuable comments. It is not uncommon in previously published reviews in Cancers that unpublished methodology of results presented in figures is shown in the figure legends; please see:
https://doi.org/10.3390/cancers13010026
https://doi.org/10.3390/cancers13010032
https://doi.org/10.3390/cancers12010254
Our review contains one figure with data not published earlier. The Reviewer criticizes this figure for not providing any methodology. We find it hard to agree with this statement. Figure legend carefully describes the methodology and technical details of the experiment evaluating the effect of added HIV proteins on transcriptions of E6 and E7 genes of HPV-16 integrated into human epithelial cell line Ca Ski. To motivate the choice of the methods used in this experiment, we have provided a reference to the experimental paper in which this approach was successfully used to describe the effect of HIV proteins on epithelial cell (Lien K, Mayer W, Herrera R, Rosbe K, Tugizov SM. HIV-1 proteins gp120 and tat induce the epithelial-mesenchymal transition in oral and genital mucosal epithelial cells. PLoS One. 2019 Dec 23;14(12):e0226343. doi: 10.1371/journal.pone.0226343. PMID: 31869348; PMCID: PMC6927651). Quantification of RNA levels of E6 and E7 oncogenes was done using standard ddCt method with normalization to the 18s rRNA as one of the best methods to estimate relative levels of gene expression. The reference to this method was also added to the legend. With this we hope that we provided sufficient methodological details to enable reproduction of the experiment.
The Reviewer 2 mentions an excessive number of references (originally 191, after revision 202). This large number of references was needed to: (1) demonstrate the increased prevalence of cancers in PLWH on ART; (2) to present evidence that this high prevalence is not purely due to the immune dysfunction; (3) to advance the concept of malignant transformation of the epithelial cells by HIV-1 proteins due to the capacity of these proteins to induce oxidative stress and to be secreted, linking evidence that the same HIV proteins induce reactive oxygen species (ROS), are exported from infected cells and possess carcinogenic properties. Altogether, the review has 191 references, after the revision the number increased to 202. This is not an exceptionally high numbers compared to reviews previously published in Cancers. As examples review by Michiko Ichii and Naoki Hosen contains 172 (https://doi.org/10.3390/cancers13010025), Borden K. L. B. 180 (https://www.mdpi.com/2072-6694/13/1/42); Taylor A. H. et al, 185 (https://www.mdpi.com/2072-6694/13/1/37), Aghakhani S. et al 230 (https://www.mdpi.com/2072-6694/13/1/35) references.
The Reviewer 2 criticizes the manuscript for the repetitiveness. There are repetitive elements in the structure of the text and these were made on purpose. Firstly, we listed five HIV proteins with carcinogenic effects. Secondly, we described that the same five proteins (presented in the same order) induce ROS. Thirdly, we demonstrate that the same five proteins are found in the extracellular space. Prior to the present format of the review, we had tried to structure this information by individual proteins; however this format was even more repetitive, since for each protein we would have to repeat that it is carcinogenic, induces ROS and is found in the extracellular space. We hope that the Reviewer will accept this explanation and the current structure of the Review.
We agree with the Reviewer 2 that a clear and concise crystallization of the main message of the review needs to be presented. To meet this request we have complemented the Review with six bullet-points.
We hope that Reviewer 2 will be satisfied with the revisions and our explanations.
Reviewer 3 Report
This review article summarizes the mechanisms of cancer development in HIV-1-infected patients. It shows wide range of the topic, but scientific depth of the review is shallow. For example, there are huge references concerning cancer development by HIV-1 Tat protein, and one can write a review article only about this issue. I feel that the authors briefly summarize past papers, and I do not understand what the authors want to say from this review.
They should mention which mechanisms are critical for cancer development in HIV-1-infected patients compared to others, hypothesize novel mechanisms of the cancer development in HIV-1-infected patients, and/or propose strategies to prevent the cancer.
In page 5, line 186-204, the authors mention about infection of epithelial cells by HIV-1. I would like to introduce a paper showing that pseudotyping of HIV-1 with an endogenous retrovirus Env protein (syncytin) occurs in infected patients (Y. Tang et al. Endogenous retroviral envelope syncytin induces HIV-1 spreading and establishes HIV reservoirs in placenta. Cell Rep. 30, 4528-4539 (2020)).
Author Response
We would like to cordially thank the Reviewer 3 for the criticisms and valuable suggestions. Indeed, there is a huge bulk of data concerning carcinogenicity of HIV Tat and there are several reviews on this topic (for example see Nunnari G, Smith JA, Daniel R. HIV-1 Tat and AIDS-associated cancer: targeting the cellular anti-cancer barrier? J Exp Clin Cancer Res. 2008 May 15;27(1):3. doi: 10.1186/1756-9966-27-3. PMID: 18577246; PMCID: PMC2438332). Therefore, we have chosen to just briefly summarize these papers. The main explanation for the carcinogenicity of Tat is its capacity to regulate transcription (Huynh D, Vincan E, Mantamadiotis T, Purcell D, Chan CK, Ramsay R. Oncogenic properties of HIV-Tat in colorectal cancer cells. Curr HIV Res. 2007 Jul;5(4):403-9. doi: 10.2174/157016207781023974. PMID: 17627503) and this cumulative mechanism is well known and we have chosen to not report it in detail. Both Reviewer 3 and Reviewer 1 mentioned the necessity to described mechanisms which are critical for cancer development in HIV-1-infected patients, specifically those on ART, as they are different in the development of cancers in untreated patients. To meet this critique, we have included in the introduction the main accepted mechanism of HIV-associated carcinogenesis (page 2 lines 60-70) with necessary references and also listed this mechanism as the main one in the summary (page 15 lines 36-38).
The whole Review actually advances a new mechanism of HIV-associated carcinogenesis in PLWH on ART. We have presented the critical steps of this mechanism as bullet-points (page 1, lines 31-46).
This mechanism implies exiting opportunities to prevent development of cancer in PLWH on ART by inhibition of oxidative stress. We found very interesting publication of this topic and would be happy to include them in the Review, if the Reviewer deems them necessary. However, the numbers of the references in the Review is 202 and addition of this section would contradict the request of the Reviewer 2, who wanted to shorten the Review and shorten the number of references. This issue deserves a separate review. We hope that the Reviewer 3 will accept this explanation.
We are very grateful to Reviewer 3 for brilliant explanation of the mechanism by which HIV enters the epithelial cells by psudotyping. We have included this mechanism to the respective section of the Review page 5 lines 216-230: “However, a “real” infection can take place as well. HIV-1 was shown to hijack other viral Envs to directly enter CD4-negative cells through pseudotyping [Aiken C. Pseudotyping human immunodeficiency virus type 1 (HIV-1) by the glycoprotein of vesicular stomatitis virus targets HIV-1 entry to an endocytic pathway and suppresses both the requirement for Nef and the sensitivity to cyclosporin A.J. Virol. 1997; 71: 5871-5877] [King B., Daly J. Pseudotypes: your flexible friends. Future Microbiol. 2014; 9: 135-137] [Tang Y., George A, Nouvet F, Sweet S, Emeagwali N, Taylor HE, Simmons G, Hildreth JE. Infection of female primary lower genital tract epithelial cells after natural pseudotyping of HIV-1: possible implications for sexual transmission of HIV-1. PLoS ONE. 2014; 9: e101367. Lately Tang Y. et al have shown that HIV-1 infected T cells can fuse to and transfer the virus to placental trophoblasts, if the later express on their surfaces the envelope glycoprotein of human endogenous retrovirus family W1, syncytin [82]. This leads to the formation of an HIV-1 reservoir in the epithelial cells [82]. Syncytin-1 derives from a family of endogenous retroviruses and originates from HERVW1 infection of human germ cells [83]. Expression of syncytin could be a common feature of an epithelial cell which make them susceptible to HIV-1 via a ”non-canonical” route of HIV-1 infection..”
Round 2
Reviewer 2 Report
The authors responded adequately to the queries.